# A Simple Method of Synthesis of 3-Carboxy-2,2,5,5-Tetraethylpyrrolidine-1-oxyl and Preparation of Reduction-Resistant Spin Labels and Probes of Pyrrolidine Series

**DOI:** 10.3390/molecules26195761

**Published:** 2021-09-23

**Authors:** Sergey A. Dobrynin, Mikhail S. Usatov, Irina F. Zhurko, Denis A. Morozov, Yuliya F. Polienko, Yurii I. Glazachev, Dmitriy A. Parkhomenko, Mikhail A. Tyumentsev, Yuri V. Gatilov, Elena I. Chernyak, Elena G. Bagryanskaya, Igor A. Kirilyuk

**Affiliations:** 1N. N. Vorozhtsov Novosibirsk Institute of Organic Chemistry SB RAS, Lavrentiev Ave. 9, Novosibirsk 630090, Russia; musatov@nioch.nsc.ru (M.S.U.); zhurko@nioch.nsc.ru (I.F.Z.); m_falcon@nioch.nsc.ru (D.A.M.); polienko@nioch.nsc.ru (Y.F.P.); parkhomenko@nioch.nsc.ru (D.A.P.); gatilov@nioch.nsc.ru (Y.V.G.); chernyak@nioch.nsc.ru (E.I.C.); egbagryanskaya@nioch.nsc.ru (E.G.B.); kirilyuk@nioch.nsc.ru (I.A.K.); 2Novosibirsk State University, Pirogova Str. 2, Novosibirsk 630090, Russia; 3Voevodsky Institute of Chemical Kinetics and Combustion SB RAS, Institutskaya 3, Novosibirsk 630090, Russia; glaza@kinetics.nsc.ru; 4Federal Research Center Institute of Cytology and Genetics SB RAS, Lavrentiev Ave. 10, Novosibirsk 630090, Russia; landselur@bionet.nsc.ru

**Keywords:** nitroxide, spin label, spin probe, ethynylmagnesium bromide, nitrone, pyrrolidine

## Abstract

Stable free radicals are widely used as molecular probes and labels in various biophysical and biomedical research applications of magnetic resonance spectroscopy and imaging. Among these radicals, sterically shielded nitroxides of pyrrolidine series demonstrate the highest stability in biological systems. Here, we suggest new convenient procedure for preparation of 3-carboxy-2,2,5,5-tetraethylpyrrolidine-1-oxyl, a reduction-resistant analog of widely used carboxy-Proxyl, from cheap commercially available reagents with the yield exceeding the most optimistic literature data. Several new spin labels and probes of 2,2,5,5-tetraethylpyrrolidine-1-oxyl series were prepared and reduction of these radicals in ascorbate solutions, mice blood and tissue homogenates was studied.

## 1. Introduction

Nitroxide spin labels are playing an important role in application of EPR spectroscopy and imaging in biophysics and structural biology providing useful information about structure and dynamics of biomolecules and mechanisms of their interaction [1,2,3]. Modern achievements in EPR technique allow for structural studies inside living cells [4,5] and visualization of biochemical processes in living tissues [6]. Regretfully, some biogenic molecules and enzymatic systems readily reduce nitroxides to diamagnetic compounds and this hinders application of nitroxide spin labels in biomedical research [7]. Bulky substituents (larger than methyl) adjacent to nitroxide group can retard nitroxides decay in biological systems. So-called sterically shielded nitroxides demonstrate much higher stability to bioreduction compared to conventional tetramethyl nitroxides [8,9]. Moreover, 3-carboxy-2,2,5,5-tetraethylpyrrolidine-1-oxyl (**1**) ([Fig molecules-26-05761-ch001]) showed slower decay in cytosolic extract than trityl radical [9]. Recently, 2,2,5,5-tetraethyl-2,5-dihydropyrrole-1-oxyls were suggested as spin labels for in-cell applications [10,11,12]. Meanwhile, sterically shielded pyrrolidine-1-oxyls show higher resistance to chemical reduction than similar 2,5-dihydropyrrole-1-oxyls [10,13,14]. In analogy to the widely used nitroxide 3-carboxy-Proxyl (**2**), **1** can be considered as a valuable precursor of various reduction-resistant spin labels. For example, the *N*-hydroxysuccinimide ester **3** has been prepared and characterized by *Rajca* et al. [14]. However, the published synthesis of **1** requires multistep procedures, which give rather low overall yield of this nitroxide. We have recently published a method of synthesis of 3,4-bis-(hydroxymethyl)-2,2,5,5-tetraethylpyrrolidine-1-oxyl (**4**) from 2-aminobutanoic acid, 3-pentanone, and dimethyl fumarate in four steps with an overall yield of 40% [15]. The previously developed strategy can be adopted for the synthesis of **1** [16]. Here, we describe how **1** can be prepared in six simple steps from cheap commercially available reagents with the yield exceeding the most optimistic literature data. A set of new reduction-resistant 2,2,5,5-tetraethylpyrrolidine-1-oxyls have been synthesized, including analogs of conventional spin labels and probes. Reduction of selected spin probes of 2,2,5,5-tetraethylpyrrolidine series in an ascorbate-containing model system, mice blood and tissue homogenates was studied.

## 2. Results and Discussion

We have recently reported that nitrone **5** can be prepared in two steps, with 52–60% yield from 2-aminobutanoic acid, 3-pentanone and dimethyl fumarate via the three-component domino process, leading to pyrrolidine **6** and following oxidation of the latter with a hydrogen peroxide-tungstate system [15]. Alkali hydrolysis of the ester groups in **5** affords dicarboxylic acid **7**, which is unstable, and similarly to β-keto-carboxylic acids, easily undergoes decarboxylation to give **8** (Figure 1). Thus, the mixture of **7** and **8** was extracted from acidified aqueous solution with ethyl acetate and the extract was heated to reflux for complete decarboxylation of **7**. Crystallization from THF gave **8** as colorless crystalline solid, the structure of this compound was unambiguously confirmed by spectral data and element analysis.

It was shown previously that treatment of 3,4-bis-(hydroxymethyl)-2,5,5-triethyl-1-pyrroline *N*-oxide with ethylmagnesium bromide does not lead to nucleophilic addition onto nitrone group, presumably due to metalation [15]. In contrast, the reactions of 2,5,5-triethyl-1-pyrroline 1-oxides with less basic vinylmagnesium bromide or allylmagnesium bromide were shown to afford corresponding nitroxides [15,17]. Here, we used even less basic organometallic reagent, i.e., ethynylmagnesium bromide. Acetylene-derived organometallic reagents were successfully used by *Hideg* et al. in their syntheses of functionalized nitroxides from substituted 1-pyrroline 1-oxides [18,19,20,21]. Nitrone **8** was treated with 10-fold excess of ethynylmagnesium bromide in THF. The conversion was complete after 24 h at ambient temperature. Acidification of the reaction mixture was necessary to extract the products after quenching with water, indicating that carboxylic group remained unaffected. The extract contained a mixture of hydroxylamines and nitroxides due to partial oxidation in aerobic conditions. To achieve complete conversion of hydroxylamine to nitroxide, air was bubbled through the methanol solution in the presence of catalytic amount of methylene blue. The resulting mixture of nitroxides could not be separated using column chromatography. Crystallization from a pentane-toluene mixture afforded yellow crystals of **10a** (Figure 2). The purity of the sample of **10a** was confirmed using HPLC and ^1^H NMR after reduction with Zn in the presence of trifluoroacetic acid in CD_3_OD (see experimental section and Appendix A).

It is known that depending on conditions, reaction of nitroxides with zinc and carboxylic acid may give hydroxylamine or amine [22]. If the reaction is carried out with excess of zinc at 60–65 °C, the nitroxide group is quantitatively reduced to secondary amine, with carboxylic, mesyloxy, amino, carboxamido, terminal ethylene and acetylene groups remaining unaffected. Reduction of nitroxide group with zinc in the presence of trifluoroacetic acid appeared to be a convenient method for identification and quantification of isomers or by-products in the samples of tetraethyl-substituted nitroxides using ^1^H NMR. The sample can be prepared for NMR analysis in 15 min from 5–15 mg of nitroxide. The resulting ammonium cation does not form an asymmetric center at the nitrogen atom, showing a single set of signals in ^1^H NMR spectra for each isomer even if another asymmetric center is present in the molecule. This method gives clear advantages over other methods, allowing for fast quantitative analysis. For comparison, reduction to hydroxylamines is reversible in neutral or basic media, e.g., upon catalytic hydrogenation or in reactions with hydrazines or hydroxylamines. Some hydroxylamines are rapidly oxidized back to nitroxide with air oxygen. Reaction of sterically shielded nitroxides with hydrazines or hydroxylamines is very slow, and it is hard to achieve complete conversion. The presence of a nitroxide in the samples leads to broadening of lines in NMR spectra and complicates the analysis. The hydroxylamines can be stabilized in acidic media, e.g., upon reduction with alcohol in strongly acidic media [23,24]; however, hydroxylammonium cation formation adds an asymmetric center to the molecule, affording a mixture of diastereomers in variable ratio with superposition of signals or broadening in the NMR spectra [25]. Some other reactions, such as conversion to alkoxyamines [26] or reduction to amines with thiols [13,27], never afford a single diamagnetic product with quantitative yield.

The ^1^H NMR spectrum of the mixture of nitroxides prepared from **8** (after reduction) showed the presence of another nitroxide, presumably the minor isomer **10b**. The ratio of isomers estimated from the NMR spectrum (see Appendix A) was 11:1. The IR spectrum of the major product indicated the presence of carboxylic group and terminal acetylene moiety. Stereochemical structure of the major product **10a** was determined using X-ray analysis of the single crystal (Appendix A). Predominant formation of **10a** obviously occurs due to a nucleophilic attack from the less hindered side of the nitrone group.

Hydrogenation of a mixture of **10a**,**b** on Pd/C in methanol leads to the formation of crystalline precipitate, presumably consisting of zwitterionic salts **12** and **13**, which are moderately soluble in methanol (Figure 3). As a result, re-oxidation of the reaction mixture with air oxygen in the presence of methylene blue usually gives mixtures of **1** and **14**, which are hard to separate using chromatography or crystallization. Careful hydrogenation of **10a,b** in methanol allowed to isolate pure **14** (major isomer), but it is hard to achieve complete conversion into **13** without partial reduction of *N*-hydroxy group to amino. Hydrogenation of **10a,b** in methanol without control of absorbed hydrogen gave **15** with nearly quantitative yield. The use of THF instead of MeOH allowed to prevent precipitation of intermediate compounds. After absorption of ca. 2.5 eq. of H_2_, re-oxidation of the reaction mixture with air oxygen in the presence of methylene blue afforded pure **1** with 80% yield. Noteworthily, we did not observe formation of **15** even upon longer hydrogenation of **10a,b** in THF. Since the conversion of **12** to **13** may be incomplete, purity of samples of **1** was always verified using ^1^H NMR after reduction with Zn/CF_3_COOH system. The overall yield of **1** from 2-aminobutanoic acid, 3-pentanone and dimethyl fumarate, thus, exceeds 19%, which is far higher than the yield described in the literature [14].

Treatment of **1** with MeOH-HCl afforded ester hydrochloride **17** (Figure 4). This compound can be prepared via reaction of **1** with diazomethane and subsequent reduction in MeOH-HCl mixture. NMR spectra of **17** revealed two sets of signals with different intensity, clearly showing formation of two diastereomeric cations **17a** and **17b**. Basification of **17** under aerobic conditions gave **16** due to rapid oxidation of the free base.

Conversion of **1** into spin label **3** capable of attachment to primary amino groups have been reported earlier [14]. Reduction of the carboxylic group in **1** opens a pathway to another set of spin labels and probes (Figure 5). Carbinol **18** was prepared via two different pathways; the best overall yield (from **8**) has been achieved using reduction of **10a,b** with LiAlH_4_ with subsequent hydrogenation and oxidation.

Appel reaction is a useful method for one-step conversion of nitroxide carbinols into corresponding bromides [28,29]. Treatment of **18** with CBr_4_ and PPh_3_ afforded **20a** with 78% yield (Figure 6). Reaction of **18** with methanesulfonyl chloride in the presence of a base gives similar yield of corresponding mesylate **20b**. Heating of **20a,b** with sodium azide in DMSO gives **21**. Reduction of azido group gave aminomethyl derivative **25**, a key compound for the synthesis of spin labels **24–27**.

The maleimide spin label **24** was prepared in a two-step procedure according to the literature protocol for similar pyrroline spin labels [11,30]. Acylation of **22** with bromoacetyl bromide or chloroacetyl chloride afforded **25** and **26**; the latter was converted into iodoacetamido spin label **27** upon treatment with NaI in acetone.

The feasibility of nitroxides for in vivo EPRI, MRI and Overhauser MRI studies have been demonstrated in numerous papers (see, for example, [31,32,33,34]). For these applications, both decay of nitroxides and their physical clearance from the tissue into the blood stream play an important role. Intracellular targeting is one of the efficient methods to increase retention in tissues [35,36]. If this technique is used, reduction-resistant nitroxides, which showed high lifetimes in cytosol extracts [9], may have an important advantage over conventional tetramethyl-substituted spin probes, which are known to undergo rapid reduction inside cells. Triphenylphosphonium group is known to provide efficient intracellular accumulation via transmembrane electrostatic potential-driven mechanism [37,38]. 2-(Triphenylphosphonio)acetamido derivatives of nitroxides were successfully used both in living tissue imaging and in treatment of oxidative stress-related pathologies [38,39,40,41]. Heating of **26** with excess of triphenylphosphine in toluene afforded reduction-resistant mitochondria-targeted nitroxide **28** (Figure 7).

Moderately basic amino groups can also provide higher retention/intracellular accumulation due to transmembrane potential or ion-trapping effect [42]. Some nitroxides with tertiary amino groups showed high efficacy as both MRI contrast agents and radioprotectors [34,43]. In this work, we tried several different methods for the preparation of reduction-resistant nitroxides with tertiary amino groups. Spin label **27** readily reacts with pyrrolidine to give **29** with nearly quantitative yield. Heating of **22** with 1.2 excess of dibromopentane gave a mixture, and a major product **30** was isolated with 60% yield. An Eschweiler–Clarke reaction afforded **31** with 70% yield.

Parameters of EPR spectra, partition coefficients octanol/water and rate constants of reduction with ascorbate of nitroxides **1**, **22** and **28–30** are listed in the Table 1. Kinetics measurements were performed in buffer at pH 7.4. Spectra recordings and partition coefficient measurements were performed in distilled water. The EPR spectra of all the nitroxides are represented by broadened triplet of doublets with additional large hyperfine splitting constant (hfc) of 0.26–0.3 mT on one of methylene hydrogens (Figure 1, cf. [14,15,44,45]).

Partition coefficients of the spin probes, defined as a ratio of concentrations in octanol to water *K_p_*, vary from 5 to 70, indicating that the nitroxides may be capable of permeating cells. The new spin probes showed rather high resistance to reduction with ascorbic acid, and the rate constants *k_red_* were similar to that of **1** [9,14].

Low rates of reduction of 2,2,5,5-tetraethyl-substituted pyrrolidine nitroxides with biogenic reductants make them promising probes for biomedical research using magnetic resonance techniques. However, enzymatic reduction of nitroxides inside cells can make a major contribution to their decay in tissues; therefore, the relative stability of the spin probes in tissues may differ from that in ascorbate system [7]. To estimate the stability of new spin probes in living systems, we investigated their decay in blood and homogenates of brain, kidney and liver. The concentration of biogenic reductants in blood plasma is relatively low, not exceeding 100 μM for ascorbate [46] and GSH [47]. The main cellular component of blood, erythrocytes, is characterized by low metabolic activity and does not contain mitochondria. The kidney and liver are the main organs that perform removal of exogenous substances from blood, and this function requires high content of redox enzymatic systems and cellular antioxidants. Moreover, the liver is extremely rich in mitochondria due to its critical metabolic function in the body, with each hepatocyte containing 1000–2000 mitochondria [48], while mitochondrion content in different parts of the kidney varies within a broad range [49]. The brain (central nervous system) has an extraordinarily high metabolic rate, because neurons need large amounts of ATP for the maintenance of ionic gradients across the cell membranes and for neurotransmission, and most neuronal ATP is generated by oxidative metabolism in mitochondria [50].

For this model study, we chose **22** and **28**. The former one has a basic primary amino group which mainly exists in protonated cationic form at physiological pH. This nitroxide is the most hydrophilic and should have the lowest permeability into cells and mitochondria. The latter, **28**, has somewhat higher *K_p_*, but it has lipophilic triphenylphosphonium cationic group. Similar compounds are capable of permeation through cellular membranes and accumulate in cells and mitochondria via a transmembrane potential-driven mechanism [37].

Both radicals showed high stability in blood, where the content of reducing agents is low (Figure 2). No decay of **22** was observed in 30 min, while concentration of **28** underwent ca. 10% decrease and plateaus in 10 min. A somewhat faster decay of **28** may be the result of its reaction with reductants inside the blood cells.

The tissue homogenates were prepared from organs frozen in liquid nitrogen. This method implies disruption of cellular membranes; however, small organelles, such as mitochondria, may remain intact. Therefore, the reduction of nitroxides in tissue homogenates should be caused both by cellular reductants and enzymatic systems released into solution after partial destruction of membranes, as well as by intact mitochondria.

The EPR measurements revealed a drastic difference in the observed kinetics of the nitroxides decay in homogenates of different organs. Both nitroxides expectedly showed the fastest decay in homogenates of liver and kidney, slower decay in brain and minor decay in heart muscle (Figure 2). Faster decay of **28** as compared to **22** in homogenates of brain and liver presumably result from targeted accumulation of the former in the remaining intact mitochondria. Taking into account the ascorbate content in tissues [46] and reduction rate constant shown in the Table 1, nearly 50% decay of **22** in half an hour cannot be explained by non-enzymatic nitroxide reduction. The recent study by Babić et al. showed that NADPH-dependent destruction by cytochrome P450 may be a major pathway of sterically shielded piperidine nitroxides decay in the presence of hepatic microsomal fraction [51]. The behavior observed here for **22** and **28** is in agreement with the above-mentioned finding. Indeed, cytochromes P450 are expressed in the liver and other organs, including the brain [52] and kidney [53], where they can contribute significantly to local metabolism. In contrast, muscles are known to have low level of xenobiotic metabolism activity [54]. Moreover, P450s are expressed in mitochondria too [55]; therefore, the accumulation of **28** in mitochondria may result in faster decay of this nitroxide compared to **22**.

## 3. Materials and Methods

The nitrone **5** was prepared according to the literature protocols [15]. IR spectra were acquired on an FT-IR spectrometer in KBr and are reported in wave numbers (cm^−1^). Reactions were monitored by TLC carried out using UV light 254 nm, 1% aqueous permanganate and 10% solution of phosphomolibdic acid in ethanol and/or Dragendorff reagent as visualizing agents. Column chromatography was performed on silica gel 60 (70−230 mesh). ^1^H NMR spectra were recorded at 300, 400 or 500 MHz, and ^13^C NMR spectra were recorded at 75, 100 or 125 MHz. ^1^H and ^13^C chemical shifts (δ) were internally referenced to the residual solvent peak. For analysis and structure assignments of nitroxides, the samples were subjected to reduction with the zinc/CF_3_COOH system before recording of the NMR spectra as described below.

### 3.1. Preparation of samples for ^1^H NMR from nitroxides

A suspension of zinc dust (100 mg) in a solution of a nitroxide (10–15 mg) in CD_3_OD (0.4 mL) in a small glass vial was heated to reflux upon vigorous stirring and trifluoroacetic acid (0.1 mL) was added dropwise. The mixture was stirred for 10–15 min and the solution was transferred into an NMR tube through a pipette tip with tightly inserted paper filter. The vial was rinsed with a small portion of CD_3_OD or CDCl_3_, and this solution was filtered into the same NMR tube until the normal NMR sample volume was reached.

### 3.2. Synthesis


*3-Carboxy-2,2,5-triethyl-3,4-dihydro-2H-pyrrole 1-oxide (**8**)*


A 4 M solution of sodium hydroxide in water (10 mL) was added to a solution of nitrone **5** (2.85 g, 10 mmol) in methanol (15 mL). The mixture was allowed to stand at room temperature for 24 h. Then, methanol was evaporated in vacuum and water residue was cooled to 5 °C. The cold solution was neutralized with cold 1M solution of sulfuric acid in water (20 mL) and extracted with ethyl acetate (3 × 10 mL). The organic phase was dried with Na_2_SO_4_ and filtered off. The yellow solution was heated to reflux under argon for 2 h (TLC control on silica gel, ethyl acetate/methanol/acetic acid—100/10/1). After that, the red solution was allowed to stand at -20 °C for 24 h. The pink crystalline precipitate of nitrone **8** was collected (1.40 g, 65%) and used for the next step without purification. Nitrone **8** was recrystallized from THF for characterization: colorless crystals, m.p. 145–146 °C. IR (KBr) *ν_max_*: 1697 (C=O). ^1^H NMR (400 MHz; CDCl_3_, δ): 0.70 (t, *J_t_* = 7.4 Hz, 3H), 0.83 (t, *J_t_* = 7.4 Hz, 3H), 1.08 (t, *J_t_* = 7.7 Hz, 3H), 1.72–1.90 (m, 3H), 1.99 (dt, *J_d_*= 14.3 Hz, *J_t_* = 7.4 Hz, 1H), 2.49 (dt, *J_d_*= 15.8 Hz, *J_t_* = 7.7 Hz, 1H), 2.64 (dt, *J_d_* = 15.8 Hz, *J_t_* = 7.7 Hz, 1H), 2.66 (dd, *J_d1_* = 18.7 Hz, *J_d2_* = 9.6 Hz, 1H), 3.06 (dd, *J_d1_* = 18.7 Hz, *J_d2_* = 8.9 Hz, 1H), 3.19 (dd, *J_d1_*= 9.6 Hz, *J_d2_*= 8.9 Hz, 1H), 13.26 (br, 1H). ^13^C{^1^H} NMR (100 MHz, CDCl_3_, δ): 7.3, 7.9, 9.3, 20.2, 27.9, 29.5, 31.4, 41.4, 81.8, 155.0, 171.9. Anal. Calcd for C_11_H_19_NO_3_: C, 61.95; H, 8.98; N, 6.57. Found: C, 62.19; H, 8.65; N, 6.62.


*3-Carboxy-5-ethynyl-2,2,5-triethylpyrrolidine-1-oxyls (**10a** and **10b**)*


A powder of nitrone **8** (10.6 g, 50 mmol) was added to a 0.5–1 M solution of ethynylmagnesium bromide in THF (500 mL) upon stirring. The mixture was allowed to stand at room temperature for 24 h (TLC control on silica gel, ethyl acetate/methanol/acetic acid—100/10/1), then quenched with water and acidified with 1 M solution of sulfuric acid in water (250–500 mL) to pH 3–4. Then, organic phase was washed with brine and dried with Na_2_SO_4_ and filtered off. The THF was distilled off in vacuum and residue was dissolved in methanol (200 mL). A 1 M solution of sodium hydroxide in water (60 mL) was added to the mixture. Then, the methylene blue (6 mg, 0.02 mmol) was added to the mixture and the air was bubbled until the solution turned dark blue. The methanol was distilled off in vacuum, while the remaining aqueous solution was washed with ethyl acetate, acidified with 1 M solution of sulfuric acid in water (32 mL) to pH < 4 and extracted with ethyl acetate (3 × 50 mL). The organic phase was dried with Na_2_SO_4_ and the solvent was evaporated in vacuum. The residue was triturated with diethyl ether, and yellowish crystalline precipitate of **10a,b** was collected, 7.3 g (62%) of orange crystals, m.p. 100–108 °C. IR (KBr) *ν_max_*: 3263 (≡C-H), 1730 (C=O), 2112 (C≡CH). Anal. Calcd for C_13_H_20_NO_3_: C, 65.52; H, 8.46; N, 5.88. Found: C, 65.66; H, 8.53; N, 5.85. ^1^H NMR (400 MHz; CD_3_OD/CDCl_3_, Zn/CF_3_COOH system δ): **10a**: 0.97 (t, *J_t_* = 7.4 Hz, 3H), 1.06 (t, *J_t_* = 7.4 Hz, 3H), 1.16 (t, *J_t_* = 7.3 Hz, 3H), 1.81 (dq, *J_d_* = 14.8 Hz, *J_q_* = 7.4 Hz, 1H), 1.86 (dq, *J_d_* = 14.8 Hz, *J_q_* = 7.4 Hz, 1H), 2.02–2.17(m, 3H), 2.24 (dq, *J_d_* = 7.3 Hz, *J_q_* = 14.2 Hz, 1H), 2.44 (dd, *J_d1_* = 13.9 Hz, *J_d2_* = 8.9 Hz, 1H), 2.61 (dd, *J_d1_* = 13.9 Hz, *J_d2_* = 7.5 Hz, 1H), 3.25 (s, 1H), 3.37 (dd, *J_d1_* = 8.9 Hz, *J_d2_* = 7.5 Hz, 1H); **10b**: 2.34 (dd, *J_d1_* = 13.7 Hz, *J_d2_*= 6.6 Hz, 1H), 2.85 (dd, *J_d1_* = 13.7 Hz, *J_d2_* = 10.5 Hz, 1H), 3.07 (s, 1H), 3.07 (dd, *J_d1_* = 10.5, *J_d2_* = 6.6 Hz, 1H), ratio of **10a**:**10b**—11:1.The sample of pure **10a** for X-ray analysis was prepared via slow condensation of pentane in toluene solution of **10a,b** at 5 °C.


*3-Carboxy-2,2,5,5-tetraethylpyrrolidine-1-oxyl (**1**)*


The hydrogenation was performed in analogy to the earlier described procedure [12]. A solution of **10a,b** (2.38 g, 10 mmol) in THF (10 mL) was placed in the reaction vessel equipped with magnetic stirrer and connection line to gasometer filled with hydrogen. The catalyst (Pd/C, 4%, 50 mg) was added and the system was purged with hydrogen and closed. The mixture was vigorously stirred until hydrogen absorption ceased (ca. 5 h, 0.6 L of hydrogen absorbed), after which the catalyst was filtered off and washed with THF. The solution was diluted with methanol (20 mL) and 1 M solution of sodium hydroxide in water (11 mL) was added to the mixture. The methylene blue (6 mg, 0.02 mmol) was added to the solution and the air was bubbled until the solution turned dark blue. The methanol was distilled off in vacuum and the remaining aqueous solution was washed with ethyl acetate. The water phase was then acidified to pH < 4 with 1 M solution of sulfuric acid in water (6 mL) and extracted with ethyl acetate (3 × 30 mL). The organic phase was dried with Na_2_SO_4_, the solvent was evaporated in vacuum and the residue was triturated with diethyl ether to give yellowish crystalline precipitate of **1** (1.9 g, 80%), orange crystals, m.p. 100–103 °C (lit. 103–106 °C [11]). IR (KBr) *ν_max_*: 1709 (C=O). Anal. Calcd for C_13_H_24_NO_3_: C, 64.43; H, 9.98; N, 5.78. Found: C, 64.41; H, 9.88; N, 5.78. ^1^H NMR (400 MHz; CD_3_OD/CDCl_3_, Zn/CF_3_COOH system δ): 0.96 (t, *J_t_* = 7.4 Hz, 3H), 0.98 (t, *J_t_* = 7.4 Hz, 3H), 0.99 (t, *J_t_* = 7.4 Hz, 3H), 1.05 (t, *Jt* = 7.4 Hz, 3H), 1.72 (dq, *J_d_* = 14.4 Hz, *J_q_* = 7.4 Hz, 1H), 1.78–2.09 (m, 7H), 2.26 (dd, *J_d1_* = 14.2 Hz, *J_d2_* = 8.1 Hz, 1H), 2.30 (dd, *J_d1_* = 14.2 Hz, *J_d2_* = 9.3 Hz, 1H), 3.12 (dd, *J_d1_*= 8.1 Hz, *J_d2_* = 9.3 Hz, 1H).


*3-Carboxy-5-vinyl-2,2,5-triethylpyrrolidine-1-oxyl (**14**)*


Isolated after hydrogenation in methanol for 3 h (ca. 60% of hydrogen absorbed) as described above for **1**. **14**: m.p. 80–83 °C, Anal. Calcd for C_13_H_22_NO_3_: C, 64.97; H, 9.23; N, 5.83. Found: C, 65.15; H, 9.34; N, 5.66. IR (KBr) *ν_max_*: 1731, 1709 (C=O); 3084 (H-C = ); 1637 (C=C). ^1^H NMR (400 MHz; CD_3_OD/CDCl_3_, Zn/CF_3_COOH system δ): 0.77 (t, *J_t_* = 7.4 Hz, 3H), 0.84 (t, *J_t_* = 7.4 Hz, 3H), 0.85 (t, *J_t_* = 7.4 Hz, 3H), 1.61–2.06 (m, 6H), 2.21 (dd, *J_d1_* = 14.2 Hz, *J_d2_* = 4.5 Hz, 1H), 2.56 (dd, *J_d1_* = 14.2 Hz, *J_d2_* = 8.0 Hz, 1H), 3.06 (dd, *J_d1_* = 8.0 Hz, *J_d2_* = 4.5 Hz, 1H), 5.28 (d, *J_d_* = 17.4 Hz, 1H), 5.36 (d, *J_d_* = 10.8 Hz, 1H), 6.00 (dd, *J_d1_* = 17.4 Hz, *J_d2_*= 10.8 Hz, 1H).


*3-Carboxy-2,2,5,5-tetraethylpyrrolidine (**15**)*


A solution of **10a,b** (0.238 g, 1 mmol) in methanol (3 mL) was placed in the reaction vessel equipped with a magnetic stirrer and connection line to gasometer filled with hydrogen. The catalyst (Pd/C, 4%, 200 mg) was added and the system was purged with hydrogen and closed. The mixture was vigorously stirred until hydrogen absorption ceased (ca. 6 h). The white crystalline precipitate formed and then disappeared in the reaction mixture during the hydrogenation. The catalyst was filtered off and washed with methanol, the solvent was distilled off in vacuum affording **15**, colorless crystalline powder, 215 mg (95%), m.p. 113–114 °C (EtOH-Et_2_O); IR (KBr) *ν_max_*: 3293, 3151 (hydrogen bonded NH, COOH), 1633, 1578 (C=O). Anal. Calcd for C_13_H_25_NO_2_×H_2_O: C, 63.64; H, 11.09; N, 5.71. Found: C, 63.38; H, 10.97; N, 5.67. ^1^H NMR (300 MHz; CD_3_OD/CDCl_3_, δ): 0.75–0.96 (m, 12H), 1.47–1.61 (m, 1H), 1.63–1.90 (m, 6H), 1.91–2.12 (m, 3H), 2.74 (t, *J_t_* = 7.2 Hz, 1H). ^13^C{^1^H} NMR (125 MHz; CD_3_OD/CDCl_3_, δ): 7.6, 7.9, 8.0, 8.1, 24.8, 28.0, 28.7, 29.5, 38.7, 52.8, 69.0, 72.5, 176.4.


*3-Methoxycarbonyl-2,2,5,5-tetraethylpyrrolidine-1-oxyl (**16**)*


The solution of CH_2_N_2_ in Et_2_O (20 mL) was prepared form *N*-nitroso-*N*-methylurea (1.16 g, 11.3 mmol), H_2_O (0.5 mL) and KOH (0.95 g, 17 mmol) and added to the solution of **1** (1.36 g, 5.6 mmol) in dry Et_2_O (20 mL). The mixture was stirred at ambient temperature for 5 h. The solvent was evaporated in vacuum and residue was separated using column chromatography on silica gel (hexane:EtOAc—9:1, R_f_ = 0.35), yielding **16**, 1.25 g (87%) orange oil. UV (EtOH) λ_max_ (log*ε*): 237 (4.94). IR (neat) *ν_max_*: 1740 (C=O). Anal. Calcd for C_14_H_26_NO_3_: C, 65.59; H, 10.22; N, 5.46. Found: C, 65.50; H, 10.27; N, 5.46.


*1-Hydroxy-3-methoxycarbonyl-2,2,5,5-tetraethylpyrrolidine hydrochloride (**17**)*


*Method A (from **16**).* To the solution of **16** (170 mg, 0.66 mmol) in MeOH (1 mL), the saturated solution of dry HCl in MeOH (4 mL) was added in one portion. The mixture was allowed to stand at room temperature for 24 h until discoloration. The solvent was evaporated in vacuum and the residue was triturated with dry Et_2_O (5 mL). The colorless precipitate of **17** was filtered off (130 mg, 67%) and recrystallized from isopropanol, m.p. 148–150 °C (isopropanol). IR (KBr) *ν_max_*: 1737 (C=O). Anal. Calcd for C_14_H_28_ClNO_3_: C, 57.23; H, 9.61; Cl, 12.06; N, 4.77. Found: C, 57.09; H, 9.81; N, 4.86; Cl, 11.92. ^1^H NMR (300 MHz; CD_3_OD + CDCl_3_, δ): 0.83 (t, *J_t_* = 7.4 Hz, 3H), 0.85 (t, *J_t_* = 7.4 Hz, 3H), 0.87 (t, *J_t_* = 7.4 Hz, 3H), 0.93 (t, *J_t_* = 7.4 Hz, 3H), 1.28 (dq, *J_d_* = 13.2 Hz, *J_q_* = 7.4 Hz, 1H), 1.04–1.40 (m, 7H), 1.85 (dd, *J_d1_* = 13.3 Hz, *J_d2_* = 7.3 Hz, 1H), 2.07 (dd, *J_d1_* = 13.3 Hz, *J_d2_* = 11.5 Hz, 1H), 2.09 (dd, *J_d1_* = 11.5 Hz, *J_d2_* = 7.3 Hz, 1H), 3.67 (s, 3H). ^13^C{^1^H} NMR (75 MHz; CDCl_3_, δ): 7.8, 7.9, 8.1, 8.3, 8.4, 21.2, 24.2, 25.3, 25.8, 27.2, 27.9, 28.4, 28.5; 33.9, 34.1, 45.2, 47.5; 52.7, 52.6, 77.7, 77.8 79.7, 80.8, 169.1, 170.2.

*Method B (from **1**).* To the solution of **1** (500 mg, 2.06 mmol) in MeOH (1 mL), the saturated solution of dry HCl in MeOH (7 mL) was added in one portion. The mixture was allowed to stand at room temperature for 72 h. The solvent was evaporated in vacuum and the residue was triturated with dry Et_2_O (5 mL). The colorless precipitate of **17** was filtered off, yield 550 mg (91%).


*3-Hydroxymethyl-5-ethinyl-2,2,5-triethylpyrrolidine-1-oxyl (**19a**,**b**) mixture of isomers*


The solution of **10a,b** (2.38 g, 10 mmol) in THF (10 mL) was added dropwise to a stirred solution of LiAlH_4_ (0.57 g, 15 mmol) in THF (10 mL). The mixture was stirred at ambient temperature for 1 h, and then the flask was cooled to 5–10 °C in a cold-water bath and quenched with water. The organic phase was separated via decantation, and the remaining viscous suspension was washed with THF (3 × 50 mL). The extract was dried with Na_2_CO_3_ and filtered. The solvent was evaporated in vacuum and the residue was triturated with hexane-diethyl ether mixture and orange crystalline precipitate of **19a,b** was collected, 2.12g (95%) of orange crystal, m.p. 76–77 °C (dec). IR (KBr) *ν_max_*: 3465 (O–H), 3242 (≡C-H), 2108 (C≡CH). Anal. Calcd for C_13_H_22_NO_2_: C, 69.61; H, 9.89; N, 6.24. Found: C, 69.33; H, 9.44; N, 5.95. ^1^H NMR (500 MHz; CD_3_OD/CDCl_3_, Zn/CF_3_COOH system δ): 0.93 (t, *J_t_* = 7.4 Hz, 3H), 0.95 (t, *J_t_* = 7.2 Hz, 3H), 1.11 (t, *J_t_* = 7.4 Hz, 3H), 1.62–1.75 (m, 3H), 1.82–1.97 (m, 4H), 2.57 (tt, *J_t1_* = 5.1 Hz, *J_t2_* = 7.6 Hz, 1H), 2.62 (dd, *J_d1_* = 12.9 Hz, *J_d2_* = 7.6 Hz, 1H), 2.88 (s, 1H), 3.15 (d, *J_d_* = 5.1 Hz, 2H).


*3-Hydroxymethyl-2,2,5,5-tetraethylpyrrolidine-1-oxyl (**18**)*


*Method A.* Reduction of **1** with LiAlH_4_ as described above afforded (**18**), yield 90%, orange oil, IR (neat) *ν_max_*: 3425 (O-H). Anal. Calcd for C_13_H_26_NO_2_: C, 68.38; H, 11.48; N, 6.13. Found: C, 68.37; H, 11.55; N, 6.45. ^1^H NMR (300 MHz; CD_3_OD/CDCl_3_, Zn/CF_3_COOH system δ): 0.92 (t, *J_t_* = 7.4 Hz, 3H), 0.93 (t, *J_t_* = 7.4 Hz, 3H), 0.95 (t, *J_t_* = 7.4 Hz, 3H), 0.96 (t, *J_t_* = 7.4 Hz, 3H), 1.64–1.95 (m, 9H), 2.24 (dd, *J_d1_* = 13.5, *J_d2_* = 7.0 Hz, 1H), 2.41 (dddd, *J_d1_* = 10.2 Hz, *J_d2_* = 7.3 Hz, *J_d3_* = 6.9 Hz, *J_d4_* = 5.5 Hz, 1H), 3.57 (dd, *J_d1_* = 10.9 Hz, *J_d2_* = 7.3 Hz, 1H), 3.70 (dd, *J_d1_* = 10.9 Hz, *J_d2_* = 5.5 Hz, 1H).

*Method B.* Hydrogenation of **19a,b** was carried out according to the procedure described above for **1**. The mixture of the nitroxide (2.0 g, 9 mmol), catalyst (Pd/C, 4%, 50 mg) and methanol (10 mL) was stirred in hydrogen atmosphere. When hydrogen absorption ceased (ca. 0.6 L of hydrogen absorbed), the catalyst was filtered off, and the air was bubbled through the solution. The methanol was distilled off in vacuum and residue was dissolved in 50 mL of ethyl acetate. The organic phase was washed with 0.1 M sulfuric acid solution (3 × 30 mL). The organic phase was separated and was dried with Na_2_CO_3_ and filtered off. The solvent was evaporated to give **18**, 1.9 g (93%).


*3-Bromomethyl-2,2,5,5-tetraethylpyrrolidine-1-oxyl (**20a**)*


The solution of PPh_3_ (0.6 g, 2.3 mmol) in dry CH_2_Cl_2_ (2 mL) was added dropwise to a stirred cold solution of **18** (0.45 g, 2 mmol) and CBr_4_ (0.73 g, 2.2 mmol) in dry CH_2_Cl_2_ (2 mL). The mixture was stirred at ambient temperature for 1 h. The solvent was evaporated in vacuum and residue was separated using column chromatography on silica gel (hexane:EtOAc—13:1), yielding **20a**: 0.45 g (78%) of orange oil, IR (neat) *ν_max_*: 648, 594 (C-Br). Anal. Calcd for C_13_H_25_NOBr: C, 53.61; H, 8.65; N, 4.81; Br, 27.44. Found: C, 53.20; H, 8.87; N, 4.74; Br, 27.24.


*3-Mesyloxymethyl-2,2,5,5-tetraethylpyrrolidine-1-oxyl (**20b**)*


The solution of DIPEA (0.8 g, 6.2 mmol) in dry CHCl_3_ (5 mL) was added dropwise to a stirred mixture of the nitroxide **18** (1.33 g, 5.8 mmol) and MsCl (0.7 g, 6.1 mmol). The mixture was stirred at ambient temperature for 2 h, then washed with water (3 × 30 mL). The organic phase was separated and was dried with MgSO_4_ and filtered, the solvent was evaporated and residue was separated using column chromatography on silica gel (chloroform), yielding **20b**: 1.35 g (76%), orange oil, IR (neat) *ν_max_*: 1357, 1176 (S=O). Anal. Calcd for C_14_H_28_NO_4_S: C, 54.87; H, 9.21; N, 4.57; S, 10.46. Found: C, 54.40; H, 9.02; N, 4.33; S, 10.03. ^1^H NMR (300 MHz; CD_3_OD/CDCl_3_, Zn/CF_3_COOH system δ): 0.96 (t, *J_t_* = 7.4 Hz, 3H), 0.97 (t, *J_t_* = 7.4 Hz,3H), 1.00 (t, *J_t_*=7.4 Hz, 3H), 1.01 (t, *J_t_*=7.4 Hz, 3H), 1.68–2.07 (m, 9H), 2.31 (dd, *J_d1_*=13.3 Hz, *J_d2_*=6.9 Hz,1H), 2.72 (dddd, *J_d1_*=10.7 Hz, *J_d2_*=7.4 Hz, *J_d3_*=6.9 Hz, *J_d4_*=6.5 Hz, 1H), 3.11 (s, 3H), 4.24 (dd, *J_d1_*=10.0 Hz, *J_d2_*= 7.4 Hz, 1H), 4.37 (dd, *J_d1_* = 10.0 Hz, *J_d2_* = 6.5 Hz, 1H).


*3-Azidomethyl-2,2,5,5-tetraethylpyrrolidine-1-oxyl (**21**).*


The mixture of the nitroxide **20a** or **20b** (16.5 mmol), NaN_3_ (5.4 g, 82 mmol) and DMSO (100 mL) was stirred at 60 °C for 30 h. The mixture was diluted with water (100 mL) and was extracted with hexane (3 × 50 mL). The organic phase was separated and was dried with Na_2_CO_3_ and filtered off. The solvent was evaporated and residue was separated using column chromatography on silica gel (hexane:EtOAc—13:1), yielding **21**: 4.0 g (97%), orange oil, IR (neat) *ν_max_*: 2098 (N_3_). Anal. Calcd for C_13_H_25_N_4_O: C, 61.63; H, 9.95; N, 22.11. Found: C, 62.10; H, 9.95; N, 22.01.


*3-Aminomethyl-2,2,5,5-tetraethylpyrrolidine-1-oxyl (**22**)*


*Method A.* The solution of nitroxide **21** (4.0 g, 16 mmol) in THF (20 mL) was added dropwise to a stirred mixture of LiAlH_4_ (0.9g, 24 mmol) in THF (50 mL). The mixture was stirred at ambient temperature for 1 h, and then the flask was placed into a cold-water bath and quenched with water. The organic phase was separated via decantation, and the remaining viscous suspension was washed with THF (3 × 30 mL); the extract was dried with Na_2_CO_3_ and filtered. The solvent was evaporated, yielding **22**, 3.24 g (90%), orange oil, IR (neat) *ν_max_*: 3377 (N-H). HRMS (EI/DFS) m/z: [M]^+^calcd for C_13_H_27_N_2_O 227.2118, found 227.2116. ^1^H NMR (300 MHz; CD_3_OD/CDCl_3_, Zn/CF_3_COOH system δ): 0.92 (t, *J_t_* = 7.4 Hz, 3H), 0.94 (t, *J_t_* = 7.4 Hz, 3H), 0.97 (t, *J_t_* = 7.4 Hz, 3H), 0.98 (t, *J_t_* = 7.4 Hz, 3H), 1.60–2.03 (m, 9H), 2.40 (dd, *J_d1_* = 13.3 Hz, *J_d2_* = 6.6 Hz, 1H), 2.66 (dddd, *J_d1_* = 11.9 Hz, *J_d2_* = 11.1 Hz, *J_d3_* = 6.6 Hz, *J_d4_* = 3.4 Hz, 1H), 2.98 (t, *J_t_* = 11.9 Hz, 1H), 3.17 (dd, *J_d1_* = 12.0 Hz, *J_d2_* = 3.4 Hz, 1H).

*Method B.* Triphenylphosphine (0.45 g, 1.7 mmol) was added to the solution of azide **21** (0.23 g, 0.9 mmol) in dry diethyl ether. The reaction mixture was stirred and heated under reflux for 2 h. The solvent was evaporated under reduced pressure and 50% aqueous ethanol (10 mL) was added to the residue. The resulted mixture was heated under reflux with stirring for 8 h. The ethanol was distilled off under reduced pressure and the water layer was acidified with 3% aqueous hydrochloric acid to pH = 3 and extracted with chloroform (5 × 5 mL). The water layer was basified with sodium carbonate to pH ~ 9–10, extracted with chloroform (5 × 5 mL), and the extract was dried with sodium carbonate, and solvent was evaporated under reduced pressure. The residue was purified by column chromatography on Al_2_O_3_ (eluent—CHCl_3_) to give **22** (0.17 g, 86%).


*3-Maleimidomethyl-2,2,5,5-tetraethylpyrrolidine-1-oxyl (**24**)*


The solution of maleic anhydride (0.2 g, 2 mmol) in chloroform (10 mL) was added dropwise to a stirred solution of the nitroxide **22** (0.34 g, 1.5 mmol) in chloroform (10 mL), and stirred at ambient temperature for 24 h. Then, the NaOAc (0.5, 6 mmol) and Ac_2_O (10.4 g, 102 mmol) were added and the mixture was stirred in an oil bath at 110 °C for 3 h. The reaction mixture was concentrated in vacuum and the residue was subjected to column chromatography on silica gel (chloroform), yielding **24**: 0.24 g (52%), orange crystals, m.p. 83–84 °C (from hexane). IR (KBr) *ν_max_*: 1701 (C=O). Anal. Calcd for C_17_H_27_N_2_O_3_: C, 66.42; H, 8.85; N, 9.11. Found: C, 66.69; H, 9.25; N, 9.08.


*3-Bromoacetamidomethyl-2,2,5,5-tetraethylpyrrolidine-1-oxyl (**25**)*


A mixture of amine **22** (0.11 g, 0.5 mmol) with DIPEA (97 μL, 0.6 mmol) in dry CHCl_3_ was cooled to 0 °C in an ice bath. After that, a solution of bromoacetyl bromide (46 μL, 0.5 mmol) in dry CHCl_3_ was added dropwise, and the reaction mixture was allowed to stand at room temperature for 3 h. The solution was carefully washed with water three times and dried with Na_2_SO_4_. The solvent was removed under reduced pressure and the residue was subjected to column chromatography on silica gel, eluent CHCl_3_ afforded **25** (0.13 g, 79%) as an orange oil. IR (neat) *ν*_max_: 1658 (C=O), 1552 (N-H). HRMS (EI/DFS) *m/z* [M]^+^calcd for C_15_H_28_BrN_2_O_2_ 347.1329, found 347.1333.


*3-Chloroacetamidomethyl-2,2,5,5-tetraethylpyrrolidine-1-oxyl (**26**)*


The solution of 2-chloroacetyl chloride (0.27 g, 2.4 mmol) in CHCl_3_ (10 mmol) was added dropwise to a stirred cold solution of the nitroxide **22** (0.50 g, 2.2 mmol) and DIPEA (0.43 g, 3.3 mmol) in CHCl_3_ (20 mL). The mixture was stirred at ambient temperature for 1 h. The mixture was washed with water (3 × 20 mL). The organic phase was separated and was dried with Na_2_SO_4_ and filtered off. The solvent was evaporated and residue was separated using column chromatography on silica gel (hexane:EtOAc—1:1), yielding **26**: 0.64 g (88%) of orange oil, IR (neat) *ν_max_*: 1666 (C=O). Anal. Calcd for C_15_H_28_N_2_O_2_Cl: C, 59.29; H, 9.29; N, 9.22; Cl, 11.67. Found: C, 59.36; H, 9.12; N, 8.75; Cl, 11.81.


*3-Iodoacetamidomethyl-2,2,5,5-tetraethylpyrrolidine-1-oxyl (**27**)*


The mixture of **26** (0.64 g, 2.1 mmol) and NaI (1 g, 6.3 mmol) in acetone (15 mL) was stirred at ambient temperature for 48 h. The precipitate was filtered off. The solvent was evaporated and the residue was diluted with water (20 mL) and extracted with diethyl ether (3 × 20 mL). The organic phase was separated and dried with MgSO_4_. The solvent was evaporated in vacuum and the residue was triturated with hexane, and precipitate of **27** was collected, 0.48 g (58%), orange crystals, m.p. 83.9 °C (dec, from diethyl ether). IR (KBr) *ν_max_*: 1643 (C=O). Anal. Calcd for C_15_H_28_N_2_O_2_I: C, 45.58; H, 7.14; N, 7.09; I, 32.10. Found: C, 45.30; H, 7.06; N, 6.85; I, 32.21. HRMS (EI/DFS) m/z: [M]^+^calcd for C_15_H_28_N_2_O_2_^127^I 395.1190, found 395.1194.


*3-(2-Triphenylphosphonio)acetamidomethyl-2,2,5,5-tetraethylpyrrolidine-1-oxyl chloride (**28**)*


A mixture of **29** (0.305 g, 1 mmol), triphenylphosphine (0.6 g, 2.3 mmol) and toluene (5 mL) was heated to reflux under nitrogen for 5 h and then left at 50 °C overnight. The solvent was distilled off in vacuum and the residue was purified using column chromatography (silica gel, eluent chloroform:ethanol—50:1). The collected fractions of **30** were concentrated in vacuum, while the residue was dissolved in a mixture of diethyl ether and benzene 1:1 and left at 10 °C for crystallization. The precipitate of **30** monohydrate was filtered off, washed with cold diethyl ether and dried. The yield was 0.240 g (41%). Pale yellowish crystals m.p. 108 °C (dec.). IR (KBr) *ν_max_*: 1670 (C=O). Anal. Calcd for C_15_H_28_N_2_O_2_I: C, 67.85; H, 7.76; N, 4.80; Cl, 6.07; P, 5.30. Found: C, 67.64; H, 8.04; N, 4.50; Cl, 6.01; P, 5.16. ^1^H NMR (after reduction with Zn/CF_3_COOH) (400 MHz; CD_3_OD-CF_3_COOH, δ): 0.80 (m, 12H), 1.54 (m, 5H), 1.68 (m, 4H), 1.95 (dd *J_d1_* = 7 Hz, *J_d2_* = 13.5 Hz, 1H), 2.27 (m, 1H), 3.00 (dd *J_d1_* = 13.5 Hz, *J_d2_* = 10.5 Hz, 1H) 3.18 (dd *J_d1_* = 13.5 Hz, *J_d2_* = 4 Hz, 1H), 4.50 (m, partly exchanged), 7.59 (m, 12H), 7.71 (m, 3H).


*3-(2-(Pyrrolidin-1-yl)acetamidomethyl)-2,2,5,5-tetraethylpyrrolidine-1-oxyl (**29**)*


A mixture of **27** (0.11 g, 0.3 mmol) and pyrrolidine (0.1 g, 1.4 mmol) in dry chloroform (10 mL) was stirred at ambient temperature for 1 h. The mixture was washed with water (3 × 20 mL). The organic phase was separated and was dried with Na_2_CO_3_ and filtered off. The solvent was evaporated and residue was separated using column chromatography on silica gel (methanol:EtOAc—1:9), yielding **29**, 0.09 g (95%) orange oil. IR (neat) *ν_max_*: 1662 (C=O). HRMS (EI/DFS) m/z: [M]^+^calcd for C_19_H_36_N_3_O_2_ 338.2802, found 338.2798. ^1^H NMR (300 MHz; CD_3_OD/CDCl_3_, Zn/CF_3_COOH system): 0.90 (t, *J_t_* = 7.4Hz, 3H), 0.92 (t, *J_t_* = 7.4 Hz, 3H), 0.97 (t, *J_t_* = 7.4 Hz, 3H), 0.99 (t, *J_t_*= 7.4 Hz, 3H), 1.62–1.97 (m, 14H), 2.38–2.54 (m, 1H), 2.94–3.25 (m, 3H), 3.39–3.51 (m, 1H), 3.59–3.81 (m, 1H), 3.97 (s, 2H).


*3-((Piperidin-1-yl)methyl)-2,2,5,5-tetraethylpyrrolidine-1-oxyl (**30**)*


The mixture of amine **22** 0.20 g, 0.9 mmol), 1,5-dibromopentane (0.24 g, 1.1 mmol) and potassium carbonate (0.62 g, 4.5 mmol) in acetonitrile (10 mL) was stirred at 60 °C. The precipitate was filtered off. The solvent was evaporated and the residue was separated using column chromatography on silica gel (hexane:EtOAc—1:1), yielding **33**, 0.15 g (60%), orange oil. HRMS (EI/DFS) m/z: [M]^+^calcd for C_18_H_35_N_2_O 295.2744, found 295.2741. IR (neat) *ν_max_*: 1461 (C-H). ^1^H NMR (300 MHz; CD_3_OD/CDCl_3_, Zn/CF_3_COOH system δ): 0.91 (t, *J_t_* = 7.4 Hz, 3H), 0.92 (t, *J_t_* = 7.4 Hz, 3H), 0.96 (t, *J_t_* = 7.4 Hz, 3H), 0.98 (t, *J_t_* = 7.4 Hz, 3H), 1.36–1.53 (br, 1H), 1.61–1.98 (m, 14H), 2.40 (dd, *J_d1_* = 13.6 Hz, *J_d2_*= 6.7 Hz, 1H), 2.70–2.92 (m, 3H), 3.10 (dd, *J_d1_* = 13.2 Hz, *J_d2_* = 2.3 Hz, 1H), 3.21 (dd, *J_d1_* = 13.2 Hz, *J_d2_* = 11.1 Hz, 1H), 3.50–3.67 (m, 2H).


*3-(Dimethylaminomethyl)-2,2,5,5-tetraethylpyrrolidine-1-oxyl (**31**)*


The mixture of amine **22** (0.15 g, 0.6 mmol), 20% formaldehyde water solution (0.5 mL, 3.3 mmol) and formic acid (0.308 g, 6.7 mmol) was stirred at 50 °C for 4 h. The mixture was diluted with sodium hydrocarbonate solution (10 mL) and was extracted with ether (3 × 10 mL). The organic phase was separated and dried with Na_2_CO_3_ and filtered off. The solvent was evaporated and residue was separated using column chromatography on silica gel (methanol:EtOAc—1:9), yielding 31, 0.12g (70%). HRMS (EI/DFS) m/z: [M]+ calcd for C_15_H_31_N_2_O 255.2431, found 255.2433. IR (neat) *ν_max_*: 1461 (C-H).HRMS (EI/DFS) m/z: [M]^+^calcd for C_15_H_31_N_2_O 255.2431, found 255.2433., ^1^H NMR (300 MHz; CD_3_OD/CDCl_3_, Zn/CF_3_COOH system δ): 0.93 (t, *J_t_* = 7.4 Hz, 3H), 0.94 (t, *J_t_* = 7.4 Hz, 3H), 0.98 (t, *J_t_* = 7.4 Hz, 3H), 1.00 (t, *J_t_* = 7.4 Hz, 3H), 1.65–2.01 (m, 9H), 2.39 (dd, *J_d1_* = 13.4 Hz, *J_d2_* = 6.3 Hz, 1H), 2.64–2.77 (m, 1H), 2.92 (s, 6H), 3.22 (dd, *J_d1_* = 12.8 Hz, *J_d2_* = 2.7 Hz, 1H), 3.32 (dd, *J_d1_* = 12.8 Hz, *J_d2_* = 11.2 Hz, 1H).

### 3.3. EPR Measurements and Kinetics

The EPR spectra in water were recorded on a Bruker ER-200D-SRC spectrometer in 50 µL glass capillary for 0.2–0.4 mM radical solutions degassed via bubbling with argon. Spectrometer settings: frequency, 9.87 GHz; microwave power, 5.0 mW; modulation amplitude, 0.05 mT; time constant, 50–100 ms; and conversion time, 5.12 ms. For kinetic measurements in water, stock solutions of nitroxide, ascorbic acid and glutathione in phosphate buffer (5 mM, pH 7.4) were prepared, and pH was adjusted to 7.4 with NaOH or HCl. All the solutions were deoxygenated with argon, carefully and quickly mixed in a small tube to attain final concentrations (nitroxide, 0.2–04 mM; GSH, 2 mM; and ascorbate, 100–300 mM) and were placed into an EPR capillary (50 μL). The capillary was sealed on both sides and placed into the EPR resonator. The decay of amplitude of the low-field component of the EPR spectrum was followed to obtain the kinetics. The initial part of the decay curves (up to 200 min) was used for fitting. Kinetics of the decay was fitted to a monoexponential function to calculate the first-order rate constants. Then, these constants were divided by the concentration of ascorbic acid to calculate the second-order reaction constants (Appendix A). Partition coefficients in a water–octanol mixture were estimated from the amplitudes of the EPR spectra of a nitroxide in water after extensive shaking with different portions of added octanol and separation using centrifugation.

The measurements of kinetics of nitroxide decay in blood and tissue homogenates were performed using an Elexsys E540 X-band spectrometer in a 50 µL glass capillary for 0.2 mM solutions, with the following spectrometer settings: frequency, 9.87 GHz; centerfield, 350.6 mT; sweep range, 10 mT; microwave power, 2.0 mW; modulation amplitude, 0.1 mT; time constant, 10.24 ms; and conversion time, 20.48 ms.

The stock solutions of the nitroxides **25** and **31** in water (1 mM) were prepared, and pH was adjusted to 7.0 with NaOH. A total of 2 μL of the nitroxide solution was added to 98 μL of the blood or tissue homogenate, giving a nitroxide concentration equal to 0.2 mM. The resulting mixture of nitroxide with blood or tissue homogenate was transferred to the glass capillary (50 μL) and the latter was placed into the EPR resonator. The decay of the double integral of the EPR spectrum was followed to obtain the kinetics.

### 3.4. Animals

Male five-month-old C57BL/6 mice were obtained from the Breeding Experimental Animal Laboratory of the Institute of Cytology and Genetics (ICG), Siberian Division of the Russian Academy of Sciences (Novosibirsk, Russia). All the experiments were carried out according to Animal Care Regulations of the Institute of Cytology and Genetics (ICG), Novosibirsk. Animals were sacrificed by decapitation, blood was collected into a microcentrifuge tube with heparin (final concentration 5%), gently rocked by hand and snap-frozen with liquid nitrogen immediately after collection. Organ tissue (brain, heart, liver and kidney) was excised on ice, rinsed with ice-cold 10 mM PBS (pH 7.4), sheared with scissors and homogenized in PBS (1/1 by weight) on ice using electric homogenizer fitted with a Teflon head. The homogenates were then additionally mixed with PBS (1/1 by weight, final dilution 1/3) to decrease viscosity for further aspiration into glass capillaries, and snap-frozen in liquid nitrogen. Before measurement, 10 mM aqueous solutions of the nitroxide probes were prepared ex tempore, adjusted to physiological pH (7.0) with NaOH and mixed with thawed homogenates (final concentration 0.2 mM). Homogenates were then loaded into 50 μL glass capillaries, which were then sealed with plasticine and inserted into the chamber of the EPR spectrometer. Time elapsed between thawing and beginning of the recording was 2 min.

## 4. Conclusions

Here, we again demonstrated the benefits of previously suggested strategy [15] for the synthesis of sterically shielded pyrrolidine nitroxides. The general approach implies assembling of 2,5,5-substituted pyrrolidine derivatives, subsequent oxidation to nitrone and addition of organometallic reagent. The use of ethynylmagnesium bromide in this reaction allowed to retain carboxylic group and to prepare 3-carboxy-2,2,5,5-tetraethylpyrrolidine-1-oxyl (**1**) with much better overall yield as compared to literature data. This method was used to prepare a set of new spin labels and probes, which showed higher resistance to reduction with ascorbate than **1**. Fast decay of the new spin probes in homogenates of liver and kidney and slow decay in homogenate of heart implies possible participation of cytochrome P450 in nitroxide decay, in analogy to the mechanism suggested by Babić et al. [51].

## 5. Patents

Part of this work is protected with a Russian patent [16].

## Data Availability

Data are contained within the article or Appendix A.

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
