# Peer review of "A Simple Method of Synthesis of 3-Carboxy-2,2,5,5-Tetraethylpyrrolidine-1-oxyl and Preparation of Reduction-Resistant Spin Labels and Probes of Pyrrolidine Series"

_molecules, 2021, doi:10.3390/molecules26195761_

Round 1
Reviewer 1 Report
This is a synthetically solid manuscript demonstrating a good approach to 5-memb. gem-diethyl nitroxides. I am a bit concerned about the reliability of authors' kinetic data on the reduction of nitroxides. In particular, based on the results on the gem-dimethyl nitroxides, presence of amino vs carboxylate group leads to faster rates of reduction with ascorbate, see refs below. Authors' results show opposite trend. Since the measured rates are quite slow, I wonder how reliable are the authors' pseudo-1st order rate const. under the conditions used? Refs: 1) Kinetics of nitroxide spin label... Magn. Reson. Imag. 1995, 13, pp. 219-226. 2) Reduction and destruction... Arch. Biochem. Biophys. 1987, 256, pp 232-243.
Author Response
We are very grateful to the reviewer for the attentive reading of the manuscript. To address his question we have repeated the measurements carefully for all the samples. The new results are given in the new version of the paper. Indeed, the reduction of the amino derivative 22 proceeded faster than that of the carboxylic acid 1, but the difference was much smaller than for tetramethyl analogs. Thе tendency and ca. 1.2-fold difference in the rate between the two compounds was also confirmed with data from another colleague, who kindly agreed to make these measurements independently. The smaller difference between the rates for tetraethyl nitroxides may result from a contribution of steric factors. We must admit that precise measurements of such small rates are not easy: the published rates for 1 are varying in a broad range and the errors are high (see [9,14]). Nevertheless, the new measurements showed similar values (for two compounds the difference between old and new values were within experimental error). As to the measurements for 1 and 22, it is most likely that the samples were mixed up during measurements because of similarity of the internal codes of the samples (DSA116 and DSA126 correspondingly). Indeed, the new value for 1 is identical to the previous one for 22.
Again, we thank the reviewer for his help to detect this confusion.
Reviewer 2 Report
Dear Ms. Modic,
Manuscript molecules-1384564 by Dobrynin and coworkers titled, "A simple method of synthesis of 3-carboxy-2,2,5,5-tetra-ethylpyrrolidine-1-oxyl and preparation of reduction-resistant spin labels and probes of pyrrolidine series", describes a new synthesis of the aforementioned compound. This synthesis is shorter than previous syntheses and gives better yields. In addition it describes the synthesis of several derivatives of this compound that can be used for bioconjugation and cellular delivery. Moreover, the reduction resistance of five of the nitroxides in the presence of ascorbate was tested and for two of the nitroxides towards bioreduction in blood and homogenates of select organs.
This is a well written paper. Given the current interest in free radicals that are resistant towards reduction for in-cell work, I recommend publication in Molecules after addressing the following minor points:
- Although the paper is very well written, it could use another round of editing. For example:
- There is a space missing between a number and a unit in many places throughout the manuscript.
- Line 44: "have been" should be "has been".
- Line 45 (and in other places in manuscript): Should write "Author et al." without commas and in italics.
- Heteroatoms in compounds should be in italics (for example in line 45).
- Line 104: remove "of"
- Line 110: ...adds AN asymmetric...
- Line 130: "gave15" should be "gave 15"
- Line 165: "later" should be "latter"
- Line 194: "hfc" should be defined
- Line 213: space inbetween "andhomogenates"
- Line 215: one half of the parenthesis is missing
- Line 254: "The observed here behavior of" should be "The behavior observed here for"
- Line 282 and in many other lines: Improper degree in °C.
- Line 284 and in many other lines: Be consistand and use "h" everywhere as an abbreviation for "hours"
- Line 386 and in many other lines: use "mL" rather than ml"
- Line 669: replace "Sshielded" with "shielded"
- Yields should be listed for all reactions in the schemes. For example Schemes 2,3 and 6 have no yields listed.
- The structures should have the same size in the manuscript.
- In Figure 2, use color code in the figure legend. The use of color is important in this figure as the shapes of the points are not easy to see in all cases. Also, it might be a good idea to show the structures of 22 and 28 on the plots.
Author Response
We thank you for your careful reading of the manuscript.
- There is a space missing between a number and a unit in many places throughout the manuscript.
Fixed
- Line 44: "have been" should be "has been".
Fixed
- Line 45 (and in other places in manuscript): Should write "Author et al." without commas and in italics.
Fixed
- Heteroatoms in compounds should be in italics (for example in line 45).
Fixed
- Line 104: remove "of"
Fixed
- Line 110: ...adds AN asymmetric...
Fixed
- Line 130: "gave15" should be "gave 15"
Fixed
- Line 165: "later" should be "latter"
Fixed
- Line 194: "hfc" should be defined
Fixed
- Line 213: space inbetween "andhomogenates"
Fixed
- Line 215: one half of the parenthesis is missing
Fixed
- Line 254: "The observed here behavior of" should be "The behavior observed here for"
Fixed
- Line 282 and in many other lines: Improper degree in °C.
Fixed
- Line 284 and in many other lines: Be consistand and use "h" everywhere as an abbreviation for "hours"
Fixed
- Line 386 and in many other lines: use "mL" rather than ml"
Fixed
- Line 669: replace "Sshielded" with "shielded"
Fixed
- Yields should be listed for all reactions in the schemes. For example Schemes 2,3 and 6 have no yields listed.
Fixed
- The structures should have the same size in the manuscript.
Fixed
- In Figure 2, use color code in the figure legend. The use of color is important in this figure as the shapes of the points are not easy to see in all cases. Also, it might be a good idea to show the structures of 22 and 28 on the plots.
We inserted colored signs into the picture legend. We also tried to insert the structures into the plots, but there is not enough space, especially on the right plot (b), the
structures must be at least twice smaller. With these smaller structures inserted the figure does not look nice.